# Efficient Cationization of Cotton for Salt-Free Dyeing by Adjusting Fiber Crystallinity through Alcohol-Water-NaOH Pretreatment

**DOI:** 10.3390/polym14245546

**Published:** 2022-12-18

**Authors:** Aini Wu, Wei Ma, Zhiyu Yang, Shufen Zhang

**Affiliations:** State Key Laboratory of Fine Chemicals, Frontier Science Center for Smart Materials, School of Chemical Engineering, Dalian University of Technology, Dalian 116023, China

**Keywords:** cationization, cotton fiber, alcohol-water-NaOH pretreatment, salt-free dyeing, crystallinity

## Abstract

Cationization of cotton is considered to be an effective way to realize salt-free dyeing of reactive dyes. However, applying cotton modified with glycidyltrimethylammonium chloride (GTA) suffers from large consumption of the cationic reagent. One of the reasons is that high crystallinity of cotton fibers hinders the penetration of the reagents into the cellulose interior and limits the reaction between them. This paper designed to use alcohol-water-NaOH system to pretreat the fibers before cationization. With this method, crystallinity of the cotton fibers is decreased and more reactive –OH is exposed, resulting in much higher fiber-reagent reactivity and increased GTA utilization. Influence of alcohol type, alcohol-to-water ratio, and quantity of NaOH on fiber crystallinity and GTA dosage for cationization are all examined. It is found that for achieving 96.0% fixation of C.I. Reactive Black 5 in the absence of salt, GTA dosage can be reduced by half when the fibers are pretreated by alcohol-water-NaOH. Compared with ethanol, *n*-propanol and isopropanol, *tert*-butyl alcohol incorporated system shows better performance in increasing fiber reactivity due to their weaker ability to dissolve ions. In this study, XRD and FT-IR are used to demonstrate changes in crystallinity of the fibers after pretreatment. The alteration in micromorphology and hydrophilicity of the pretreated fibers is observed by SEM and water contact angle test, respectively. Furthermore, the alcohol-water-NaOH system can be recycled to show very good repeatability. Notably, all dyed samples pretreated with the system present high color saturation and satisfactory color fastness, especially that the wet rub fastness reaches 4–5 grade, which is one grade higher than that obtained from the conventional dyeing with salt. The above findings prove that alcohol-water-NaOH pretreatment is effective in enhancing reactivity of the cotton fibers and penetrability of the agent, and it shows promising prospects in real application.

## 1. Introduction

The unique advantages of cotton fibers, such as softness, ease of dyeing, and biodegradability, make them prevail and widely used in the marketplace [1]. Reactive dyes, possessing bright colors, a wide color spectrum, and excellent color fastness due to the strength to react with cotton fibers, are the primary dyes for dyeing cotton fibers [2]. However, a large amount of inorganic salt, such as sodium sulphate, need to be added in reactive dyeing for promoting dye adsorption. It is worth noting that even when 30–150 g/L salt is added to conventional dyeing, 30–50% of the dye is still squandered [3]. Arbitrary discharge of dyeing wastewater holding substantial volumes of inorganic salts which are not biodegradable will generate environmental problems, such as salinization of the land and water pollution [4]. Thereby, avoiding the employment of salt and improving dye fixation are crucial to realize the environmental friendliness and efficient exploitation of resources in deep color dyeing techniques [5].

Fiber chemical modification [6,7], especially with glycidyltrimethylammonium chloride (GTA) or 3-chloro-2-hydroxypropyl-trimethylammonium chloride (CHPTAC) as cationic reagent for cationization, is the most explored method for salt-free dyeing [8,9]. The reaction mechanism of GTA with natural cotton fibers, and that between dyes and cationic cotton fibers are shown in Figure 1. The cationic fibers exhibit positive charges and show electrostatic attraction to anionic dyes in water, enabling salt-free, high fixation dyeing [10]. However, a high volume of cationic reagents exhausted during cationization hinders further scale-up of the method [11]. For example, when the dye dosage is 2% (o.w.f), the quantity of the cationic reagent required is calculated to be 0.6–0.8 g per gram fiber [12,13]. And as the dye usage rises, the cationic reagent consumed grows, signifying an increase in process costs and a decrease in the feasibility of the technique. The significant consumption of GTA is because that in addition to the reaction with fibers, GTA still undergoes a hydrolysis reaction as shown in Figure 1 [14]. Researchers have attempted to enhance the reaction efficiency by optimizing process parameters such as dosage amount, bath ratio, solvent, reaction temperature and time; however, the improvement is limited [15,16].

The crystallinity of cotton fibers is usually around 70%, which hinders the penetration of reagents into them and the following reaction with fibers [17,18]. Pretreatment of fibers before cationization, such as liquid ammonia treatment and mercerization, or ultrasound-assisted cationization, can boost the reactivity of fibers [19,20,21]. Yet, the liquid ammonia treatment has high apparatus requirements, and the issues of mercerization are the high alkali consumption (typically 220–280 g/L), which make the practical application of both difficult [22,23,24]. Literatures reported that in the synthesis of some cellulose derivatives, cellulose was first treated with alcohol-water-alkali solution for enhanced reactivity. Through treatment, crystallinity of the cellulose decreases, forming additional active centers, which facilitates the diffusion of the reagents and further reaction [25,26,27]. The advantages of this method are as follows: firstly, due to the lower solubility of alkali in organic solvents than in water, enrichment of alkali in cellulose is favored, consumption of alkali in this approach declines compared to that in mercerization; secondly, alcohol can weaken the hydrogen bonds that bind the fiber molecules together and widen the distance between cellulose molecular chains, which enhances the penetration of reagents in the subsequent reaction [28,29,30]. However, this kind of treatment has not been previously studied in cationization of cotton fibers for salt-free reactive dyeing.

This work is the first to conduct the combination of alcohol-water-NaOH pretreatment to activate fibers and cationic modification to realize efficient cationization for salt-free reactive dyeing. C.I. Reactive Black 5, a kind of widely used black dye is used for dyeing investigation. Black dye is chosen because of its large consumption in conventional dyeing process, and the corresponding salt dosage is also large. Hence, its study is of great significance to solve the traditional reactive dyeing problem [31,32]. In this research, the alcohol-water ratio, the quantity of NaOH, and the type of alcohol (ethanol, *n*-propanol, isopropanol, and *tert*-butyl alcohol) were all regulated to examine their impacts on the pretreatment effect. The selected alcohols are relatively less harmful to the environment, moreover, to minimize the influence on the environment and human health, the pretreatment temperature is chosen to be room temperature and the left pretreatment solution is recycled after the pretreatment. Furthermore, the following analyses including nitrogen content test, X-ray diffraction (XRD), Fourier transform infrared spectroscopy (FT-IR), scanning electron microscopy (SEM), and water contact angle test were employed to determine the alterations in structure and properties of the fibers. In addition, recycling of the pretreatment solution and dyeing properties were evaluated to show the applicability of this pretreatment method.

## 2. Materials and Methods

### 2.1. Materials

Bleached, desized cotton fabrics (225 g/m^2^) were obtained from Testfabric, Inc., Shanghai, China. Glycidyltrimethylammonium chloride (GTA) was purchased from Aladdin Chemical Reagent Co., Ltd., Shanghai, China. C.I. Reactive Black 5 was supplied by Jiangsu Jinji Industrial Co., Ltd., Jiangsu, China. The reagents such as sodium hydroxide, sodium sulfate, and anhydrous sodium carbonate were bought from Tianjin Bodi Chemical Co., Ltd., Tianjin, China. Ethanol, *n*-propanol, isopropanol, and *tert*-butyl alcohol were procured from Tianjin Damo Chemical Co., Ltd., Tianjin, China. They were used as received without further purification.

### 2.2. Pretreatment Process of Cotton Fibers

The pretreatment process of cellulose was described as follows. Firstly, a certain amount of sodium hydroxide was dissolved in the pre-mixed alcohol-water system by ultrasonic treatment at a concentration. Then the cotton fiber (4.0 g) was immersed in the prepared pretreated solution (liquor ratio = 10:1). Finally, the whole combination was transferred into a laboratory dyeing machine (Xinwang Dyeing & Finishing Machinery Factory, Jingjiang, China) and kept shaking for two hours at room temperature.

Cycled pretreatment experiments were carried out to examine the recyclability of the solvent utilized in the pretreatment, and the entire experiment procedure was depicted as follows. At first, a pretreatment experiment was conducted, then the pretreated fiber was removed from the pretreatment solution into the centrifuge tube separately and centrifuged at 9000 rpm/min. The collected liquid was combined with the solvent previously left in the Erlenmeyer flask and kept for use. 5 mL of the pretreatment solution was taken out for the acid-base titration to determine the amount of NaOH consumed in the pretreatment process. Based on the results of the titration experiment, NaOH and solvent were added to the pretreatment solution to maintain the initial values of the base amount and bath ratio. Then six cycled pretreatment experiments were performed, and the resulting pretreated fibers were left for cationization.

### 2.3. Preparation of Cationic Fibers

Cationization of raw and pretreated fibers (4.0 g) was performed by the impregnation method, respectively. The bath ratio was 3:1, the molar ratio of GTA to NaOH was 1.2:1, and the exhaust cationization technique was presented in Appendix A. Initially, the natural cotton fiber was immersed in GTA solution at 30 °C for 30 min. After 30 min, an amount of NaOH was added, and then the temperature was raised to 60 °C at a rate of 2 °C/min and maintained for 30 min. After the cationization, the cotton fabric was taken out and rinsed with tap water to wash off the cationic reagents that were not bound to fibers and then squeezed.

### 2.4. Dyeing Procedures

All dyeing was conducted with a 250 mL Erlenmeyer flask in a laboratory dyeing machine at a liquor-to-goods ratio of 20:1. All the samples were dyed with C.I. Reactive Black 5 at a concentration of 6% (o.w.f) and the dye structure is shown in Appendix A.

Exhaust process was adopted for traditional dyeing of natural fibers (4.0 g) (Appendix A) and non-salt dyeing for various cationized cotton. The original fiber was immersed in a dye solution containing 8.0 g of sodium sulfate at 30 °C to commence dyeing and maintained for 30 min, and then the temperature was stepped to 60 °C, at which time 1.6 g of sodium carbonate was replenished to the dye bath and held for 40 min. Except that no inorganic salts and alkalis were consumed during the procedure, the zero-salt dyeing operation was the same as the conventional dyeing procedure.

The dyed cotton fabrics were rinsed thoroughly with tap water, then soaped with 2 g/L detergent at 95 °C for 10 min, then the samples were washed with cold water and air dried.

### 2.5. Characterization

X-ray diffraction (XRD) information, analyzing the effect of pretreatment and cationization on fiber crystal structure, was acquired from an X-ray diffractometer (Rigaku D/MAX2400, Rigakub Co., Tokyo, Japan) using Cu Ka1 radiation. The Nicolet 6700 FTIR spectrometer (Thermo Electron Co., Waltham, MA, USA) was utilized to collect the Fourier-transform infrared spectroscopy of all the samples, to analyze the effect of pretreatment on fiber crystal type and crystallinity. To investigate the distinction of surface morphology among the raw fibers, pretreated fibers, and cationic fibers, microstructure images of the selected fibers were observed using scanning electron microscopy (SEM; JSM-5600LV, JEOL Co., Tokyo, Japan). The water contact angle tester (SL250, USA KINO Industry Co., Ltd., Shanghai, China) was used to illustrate the influence of the investigated pretreatment and cationization methods on the water absorption capacity of the fibers.

### 2.6. Measurements

Dye exhaustion (*E*%) and fixation (*F*%) were calculated based on Equations (1) and (2).
*E% =* (1 − *A*_1_/*A*_0_) *×* 100(1)
*F% =* (*E%* − *A*_2_/*A*_0_) *×* 100(2)
where *A*_0_, *A*_1_, and *A*_2_ were the absorbance of the dye solution before and after dyeing, and the soaping solution, respectively, and the specific values were acquired by UV-Vis spectrophotometer (Agilent HP8453, HP Co., Palo Alto, CA, USA).

Color strength (K/S) was tested by colorimeter (YS6060, Shenzhen Sanenshi Science & Technology Co., Ltd., Shenzhen, China).

Nitrogen content (*N*%) was measured by the Kjeldahl method, where the formula used was Equation (3).
*N% =* 2 × (*V*_1_ − *V*_2_) *× c ×* 14.01/*m ×* 100%(3)
where *V*_1_ and *V*_2_ (L) were the total volumes of acid before and after titration, respectively; *c* (mol/L) was the concentration of acid used for titration, 14.01 (g/mol) is the atomic weight of nitrogen, and m was the mass of the cotton fiber measured.

Rubbing fastness was tested according to ISO 105-X12:2001 with the help of a Y (B) 571-II crockmeter (Wenzhou Darong Textile Instrument Co., Wenzhou, China). Washing fastness was assessed according to ISO 105-C10:2006 using a laboratory dyeing apparatus (Xinwang Dyeing & Finishing Machinery Factory, Jingjiang, China).

## 3. Results and Discussion

### 3.1. Exploration of Pretreatment Process

In this research, we pretreated the fibers with an alcohol-water-NaOH solution, followed by cationization and salt-free dyeing, the whole process was presented in Figure 2. Specifically, considering the economic advantages and safety of ethanol, ethanol-water-NaOH solution was first selected to study its pretreatment effect, in which the ethanol-water ratio and NaOH dosage were the two investigated factors. Then, the influence of *n*-propanol, isopropanol, and *tert*-butyl alcohol incorporated systems was further explored to examine the universality of the method.

Effect of V_EtOH_:$V_{H_{2}O}$ in pretreatment on GTA usage for cationization was first explored. The ethanol content in the pretreatment solution affected pretreated fiber reactivity and, consequently, the consumption of GTA in the cationization. Therefore, it was desirable to find a suitable ethanol proportion to minimize the usage of GTA. The fibers were pretreated with ethanol-water-NaOH systems having different ethanol-water ratios from 0:10 to 10:0 and a constant NaOH dosage of 100 g/L. GTA amount of 55 g/L was selected for subsequent cationization, and the related results were displayed in Figure 3a. In Figure 3a, when V_EtOH_:$V_{H_{2}O}$ changed from 0:10 to 5:5, the increase in the volume of ethanol in the pretreatment solution raised the fixation of salt-free dyeing from 81.9% to 99.7%, and when V_EtOH_:$V_{H_{2}O}$ was between 5:5 and 9:1, the growth in ethanol did not cause a significant change in the fixation of C.I. Reactive Black 5, which all maintained a high level. Furthermore, when V_EtOH_:$V_{H_{2}O}$ was 10:0, the dye fixation decreased slightly to 99.2%. The above results indicated that the increase in ethanol volume facilitated higher fiber reactivity and was promising to reduce the GTA amount consumed in the cationization.

Then, considering that the dye fixation was constant and higher than 99.7% when V_EtOH_:$V_{H_{2}O}$ was between 5:5 and 9:1, the pretreated fibers obtained from this series of pretreatment solutions were cationized with a smaller GTA dose of 35 g/L to explore the variation of dye fixation, the outcomes were depicted in Figure 3b. It can be found in the figure the fixation was 92.4% when V_EtOH_:$V_{H_{2}O}$ was 5:5, and the dye fixation was boosted to 96.7% when V_EtOH_:$V_{H_{2}O}$ was 6:4. Then continuing to increase the proportion of ethanol, the dye fixation changed slightly, which was between 95.9% and 97.7%.

In this way, the fibers got from pretreatment systems with an ethanol-to-water ratio between 5:5 and 9:1 could achieve a good sale-free dyeing performance even when GTA dosage in the cationization was declined to 35 g/L. This is because ethanol has a weaker ability to dissolve ions than water, so when the ethanol percentage in the pretreatment system exceeded 50%, the promotion of NaOH penetration into fibers by the ethanol-water system was enhanced, making the NaOH concentration in the cellulose phase increase significantly. Eventually, improvements to the decrystallization process led to an enhancement in fiber reactivity [33].

Secondly, NaOH content in pretreatment on fixation of salt-free dyeing was investigated. In the ethanol-water-NaOH pretreatment system, the role of NaOH is to destroy the crystalline zone of the fibers. Because ethanol is weaker than water in terms of polarity and the ability to dissolve NaOH, excessive NaOH will precipitate from the system as the volume of ethanol increases, and the precipitated NaOH is hardly involved in the process of changing the crystalline zone of the fiber, so it is necessary to find a suitable amount of NaOH corresponding to the ethanol-to-water ratio. Accordingly, the minimal NaOH exhaustion and the corresponding ethanol-water ratio were explored within the ethanol-to-water ratio between 5:5 and 9:1. GTA amount required for the subsequent cationization of the pretreated fibers was set at 35 g/L, and the findings were shown in Figure 3c. When V_EtOH_:$V_{H_{2}O}$ was 5:5, NaOH addition of 90 g/L and 80 g/L were chosen to conduct the pretreatment of fibers, it can be seen in Figure 3c that the dye fixation of salt-free dyeing was 92.8% at NaOH usage of 90 g/L, while only 76.8% was at 80 g/L NaOH dosage. When V_EtOH_:$V_{H_{2}O}$ was 6:4, also using NaOH amount of 80 g/L in pretreatment, the dye fixation of salt-free dyeing reached 89.3%, which was 12.5% higher than that acquired when V_EtOH_:$V_{H_{2}O}$ was 5:5, so the increase of ethanol proportion can reduce the quantity of NaOH used for pretreatment and ensure the higher dye fixation. Similarly, when the NaOH dosage was 70 g/L, the dye fixation was 78% when V_EtOH_:$V_{H_{2}O}$ was 6:4 and 96.4% when V_EtOH_:$V_{H_{2}O}$ was 7:3. When the amount of NaOH was 60 g/L in the pretreatment system, the color fixation of salt-free dyeing increased from 81.2% to 87.1% when V_EtOH_:$V_{H_{2}O}$ changed from 7:3 to 8:2 while continuing to increase the volume of ethanol in the pretreatment system did not further increase the fixation. Hence, in the subsequent investigation exploring the type of alcohol on pretreatment effect, the alcohol-to-water ratio and the quantity of NaOH were chosen as V_alcohol_:$V_{H_{2}O}$ of 8:2 and 60 g/L of NaOH, respectively.

Thirdly, effect of alcohol type in pretreatment on F% of C.I. Reactive Black 5 in salt-free dyeing was studied. To investigate the generality of such alcohol-water-NaOH system on improving the reaction activity of fibers with GTA, besides ethanol, three other alcohols with less polarity including *n*-propanol, isopropanol, and *tert*-butyl alcohol were selected for pretreatment, and the results were shown in Figure 3d. At first, the fibers pretreated with *n*-propanol-water-NaOH, isopropanol-water-NaOH, and *tert*-butyl alcohol-water-NaOH were all modified with 35 g/L GTA, the dye fixation reached 98.3%, 98.8%, and 98.7%, respectively, which is much higher than that obtained when ethanol-water-NaOH was used. So, based on the above results, the dosage of GTA could be further decreased. When the fibers were modified with 30 g/L GTA, the corresponding dye fixation still reached 96.0%, 97.8%, and 96.1%, respectively.

The reason why the three alcohols are much more effective in enhancing fiber reactivity was mainly due to the low solubility of NaOH in the alcohols. After adding NaOH into the alcohol-water system, the corresponding pretreatment systems became two-phase systems with one phase containing a high concentration of NaOH, water, and a relatively small percentage of alcohol, so the cellulose immersed was surrounded by a high concentration of NaOH and could be better swollen, which facilitated the formation of a more reactive cellulose II, and thus the amount of NaOH needed for pretreatment and the dose of GTA applied in the cationization were declined.

As GTA contains quarternary ammonium group, the nitrogen content tests are performed on the cationic fibers for comparison. Nitrogen content of NCC, PCC(EtOH-70), and PCC(NPA-60) were measured and the results were shown in Figure 4. NCC is natural-cationic fiber, PCC(EtOH-70) and PCC(NPA-60) are cationic fibers pretreated with ethanol-water-NaOH (70 g/L), and *n*-propanol-water-NaOH (60 g/L), respectively, where V_alcohol_:$V_{H_{2}O}$ was 8:2. In Figure 4, when GTA was 30 g/L, nitrogen content of NCC, PCC(EtOH-70) and PCC(NPA-60) was 0.14%, 0.18% and 0.22%, respectively. When GTA dosage was elevated, the nitrogen content of all three fibers increased, with PCC(NPA-60) increasing the fastest, followed by PCC(EtOH-70), and NCC the slowest. When 55 g/L of GTA was used, the nitrogen content of PCC(NPA-60), PCC(EtOH-70) and NCC was 0.39%, 0.29%, and 0.17%, respectively. It can be seen that the nitrogen content of PCC(NPA-60) was more than twice of that of NCC, indicating much higher reactivity of PCC(NPA-60). In addition, it is noteworthy that when GTA dosage was 72.5 g/L, the nitrogen content of NCC was only 0.19%, which was even slightly lower than that of PCC (EtOH-70) at GTA dosage of 35 g/L and PCC (NPA-60) at GTA dosage of 30 g/L. Thereby, nitrogen content tests also proved alcohol-water-NaOH pretreatment could distinctly improve the fiber reactivity.

### 3.2. XRD Analysis of Cotton Fibers

The effects of various pretreatment systems on crystal structure and crystallinity of the fibers were analyzed by XRD to find the reasons for the increased efficiency of cationization from the changes in fiber microstructure. The influence of the ethanol-water-NaOH system with 100 g/L NaOH and various V_EtOH_:$V_{H_{2}O}$, including 0:10, 3:7, 5:5, 8:2, and 10:0, on the crystal structure and crystallinity of the fabrics were first explored by XRD, and the findings were given in Figure 5a. It can be seen that the peak positions of the natural cotton fiber (NC) and pretreated fibers (PC(EtOH-100)) are both 14.7°, 16.5°, 22.7°, and 34.4°, which are consistent with the reported characteristic peaks of cellulose I [26,34], indicating that the above pretreatment did not change the fiber crystal type. The reduction in peak sharpness reflected the decrease in fiber crystallinity. It is noteworthy that the sharpness of the diffraction peaks decreased significantly for pretreated fibers when V_EtOH_:$V_{H_{2}O}$ was between 5:5 and 8:2, and the intensity of the peaks decreased to a greater extent as the volume of ethanol increased, indicating the crystallinity of the fibers got lower with an increasing volume of ethanol. When the ethanol-to-water ratio was between 5:5 and 8:2, the presence of a large proportion of ethanol in the system favored the enrichment of NaOH in the cellulose phase and increased the distance between cellulose molecular chains by weakening the hydrogen bonds that bind the fiber molecules together, allowing more Na^+^ to enter both the amorphous and crystalline regions of the fibers and breaking the original hydrogen bonds in the crystalline zone, the result was the crystallinity of the pretreated fibers was reduced [17,33,35]. However, the sharpness of the peaks changed slightly when V_EtOH_:$V_{H_{2}O}$ was 10:0 because the NaOH concentration around the fibers immersed in the system was low [26].

Following XRD tests were conducted on the fibers pretreated with an alcohol-water-NaOH system consisting of different alcohols, including PC(EtOH-70), PC(NPA-60), PC(IPA-60), and PC(TBA-60), and the results were depicted in Figure 5b. The samples were pretreated using the alcohol-water-NaOH system of ethanol, *n*-propanol, isopropanol, and *tert*-butyl alcohol, respectively, where V_alcohol_:$V_{H_{2}O}$ was 8:2, NaOH dosage was 70 g/L for ethanol-water-NaOH and 60 g/L for the other three systems. In Figure 5b, PC(EtOH-70) maintained the same peak positions as NC at 14.7°, 16.5°, 22.4°, and 34.5°, and the sharpness of the peaks declined, indicating a decrease in PC(EtOH-70) crystallinity. However, the peak positions of PC(NPA-60), PC(IPA-60), and PC(TBA-60) shifted from 14.7°, 16.5° and 22.7° to 12°, 20.1° and 21.8°, respectively, i.e., the crystal structure of the pretreated fibers transformed from cellulose I to cellulose II. This was because *n*-propanol, isopropanol, and *tert*-butyl alcohol had a weak capacity to dissolve NaOH, almost all NaOH was dissolved in water and a high concentration alkali solution was formed. Therefore, the alcohol-water-NaOH was more destructive to the cellulose crystal structure and facilitated the development of cellulose II [29]. Compared with cellulose I, cellulose II had higher porous volume, higher surface wettability, and lower crystallinity, thereby, the reactivity of PC(NPA-60), PC(IPA-60), and PC(TBA-60) were higher than PC(EtOH-70), and the quantity of GTA needed to attain the same grafting effect was lower [36,37].

Besides, the crystallinity of the pretreated fibers mentioned in Figure 5b was obtained using MDI Jade 6 software, and the results were listed in Table 1. As can be seen in Table 1, the crystallinity of the natural fiber was 72.3%, and the crystallinity of the pretreated fibers ranged between 53.4% and 61.2%, indicating that pretreatment was beneficial to reduce the crystallinity of the fibers and improve the reactivity of the fibers and therefore decrease the usage of reagents employed in the cationization.

Further, to investigate the impact of cationization on crystal structure of cellulose, XRD tests were performed on cationic fibers including NCC, PCC(EtOH-70) and PCC(NPA-60), and comparison was made with the uncationized fibers. The analysis findings were depicted in Figure 5c. It can be seen that the crystal structures of the three cationic fibers: NCC, PCC(EtOH-70), and PCC(NPA-60) were almost the same as those of NC, PC(EtOH-70), and PC(NPA-60), individually, that is to say, the cationization almost did not affect the crystal structure of the fibers.

### 3.3. FT-IR Analysis of Cotton Fibers

Infrared spectroscopy can be used to evaluate the influence of the pretreatment on cellulose structure by assessing the alterations in position and intensity of the characteristic peaks [38]. Besides, FTIR can be implemented to calculate the changes in cellulose crystallinity semi-quantitatively based on the alteration of A_1430_/A_897_ ratio (crystallinity index CI) [34,39]. The peak at 1430 cm^−1^ is attributed to symmetric bending of CH_2_ groups (scissoring), and the peak at 897 cm^−1^ is perceived as valence vibration of C–O–C group or vibrational modes of groups at 1C position [40], which are sensitive to the variation in the crystalline and amorphous regions of cellulose, respectively. The smaller the A_1430_/A_897_ ratio is, the smaller the crystallinity is. In this work, natural cotton fiber (NC) and a series of pretreated fibers were subjected to FT-IR tests.

Figure 6a exhibits the differences in FT-IR spectra between NC and various pretreated fibers (PC(EtOH-100)) which were pretreated using the ethanol-water-NaOH systems with V_EtOH_:$V_{H_{2}O}$ from 5:5 to 9:1. In Figure 6a, the pretreated and raw fibers have the same characteristic peaks, where the weak peaks at 3335 cm^−1^ and 3270 cm^−1^ are assigned to stretching vibrations of the hydrogen-bonded 2OH⋯6OH⋯3OH⋯5O groups and stretching vibrations of the intramolecular hydrogen-bonded 2OH⋯6OH⋯3O groups, respectively, which are the typical peaks of cellulose I, and the partial enlargement of Figure 6a, which allowed a better observation of the peaks at 3335 cm^−1^ and 3270 cm^−1^, was depicted in Appendix A. Moreover, C-H stretching vibration at 2900 cm^−1^, the peaks at 1430 cm^−1^, and 897 cm^−1^ are also characteristic peaks of cellulose I. That is, the above pretreatment did not change the crystal structure of the fibers. Significantly, the peak intensity of the pretreated fibers located at 1430 cm^−1^ was weaker compared to NC, while the peak intensity at 897 cm^−1^ was increased, indicating that the pretreatment made the fibers less crystalline.

FT-IR tests were also performed on PC(EtOH-70), PC(NPA-60), PC(IPA-60), and PC(TBA-60) to compare the pretreatment effects of different systems, which were the pretreated fibers obtained from four various alcohol-water-NaOH systems of ethanol, *n*-propanol, isopropanol, and *tert*-butyl alcohol, respectively, where V_alcohol_:$V_{H_{2}O}$ was 8:2. The data were shown in Figure 6b. Comparing the infrared spectra of NC and PC(EtOH-70), it can be seen that the characteristic peaks of cellulose I were reflected in the spectra of both. However, some differences appeared on the IR spectra of the other three pretreated fibers. First, the appearance of the weak peaks at 3484 cm^−1^ and 3440 cm^−1^, which were attributed to vibration of intramolecular hydrogen-bonded –OH groups and O(2)H⋯O(6) intramolecular hydrogen bonding, respectively, implying the development of cellulose II, and the partial enlargement of Figure 6b, which allowed a better observation of the peaks at 3484 cm^−1^ and 3440 cm^−1^, was depicted in Appendix A. Besides, the peaks at 2900 cm^−1^, 1430 cm^−1^, and 897 cm^−1^ shifted to 2892 cm^−1^, 1420 cm^−1^, and 894 cm^−1^, respectively, all of which were manifestations of cellulose II formation [40,41]. Moreover, it is significant to note that the intensity of the peak at 1430 cm^−1^ or 1420 cm^−1^ decreased for all the pretreated fibers and increased at 897 cm^−1^ or 894 cm^−1^, revealing a reduction in the crystallinity of the pretreated fibers.

The crystallinity index (CI) was calculated for the raw and pretreated fibers to clearly demonstrate the effect of pretreatment on fiber crystallinity. The pretreated fabrics included PC(EtOH-70), PC(NPA-60), PC(IPA-60) and PC(TBA-60), the results were depicted in Figure 6c. The calculation formula is A_1430_/A_897_ for NC and pretreated fibers with the crystal structure of cellulose I, A_1420_/A_894_ for the pretreated fibers with the crystal structure of cellulose II [39]. For convenience, it is written uniformly as A_1430,1420_/A_897,894_. In Figure 6c, for NC, A_1430,1420_/A_897,894_ was 0.65, and for the pretreated fibers, A_1430,1420_/A_897,894_ was between 0.41 and 0.54, showing that pretreatment reduced the crystallinity of the fiber, which was the same as the results obtained by XRD tests, that is, the crystallinity of the pretreated fibers was lower than that of NC, indicating that the accessibility of the pretreated fibers was improved.

### 3.4. SEM Analysis of Surface Morphology of Cotton Fibers

The effect of pretreatment and cationization on the fiber surface morphology can be revealed by SEM. The SEMs of natural fibers (NC), pretreated fibers including PC(EtOH-70), PC(NPA-60), and cationic fibers including NCC, PCC(EtOH-70), and PCC(NPA-60) were shown in Figure 7a–f, respectively. It can be seen from Figure 7a that the original fiber shape is like a flat ribbon. After pretreatment with ethanol-water-NaOH or *n*-propanol-water-NaOH, the pretreated fibers, i.e., PC(EtOH-70), PC(NPA-60), were swollen and changed to a nearly cylindrical shape, as shown in Figure 7b–d shows the surface morphology of NCC, which underwent only slight swelling, without obvious difference in morphology compared to NC. Figure 7e,f depict the morphology of PCC(EtOH-70) and PCC(NPA-60), which also showed swollen and smoother morphology compared with NCC, and quite similar one compared with PC(EtOH-70) and PC(NPA-60). The above results demonstrate that pretreatment rather than cationization had a significant influence on fiber morphology. After pretreatment, the fibers took on a nearly cylindrical shape instead of a flat ribbon, allowing it to reflect light more evenly from all sides and making the fiber surface more lustrous, which will be beneficial for obtaining darker color in further dyeing process.

### 3.5. Water Contact Angle Analysis of Cotton Fibers

Water absorption of fiber was related to the percentage of the amorphous zone of the fiber and the quantity of available hydroxyl groups contained in cellulose. The more amorphous the fabric is and the more -OH groups are accessible, the better it is for absorbing moisture [41]. Cotton fiber is a porous and hydrophilic material, water contact angle can reflect the water absorption capacity of the material under test. In this study, water contact angle test was performed on both untreated and pretreated cotton fibers, and the water absorption capacity of the fiber was judged according to water contact angles at different time and the time required for complete water droplet penetration [42]. The results of water contact angle tests on NC, PC(EtOH-70), PC(NPA-60), NCC, PCC(EtOH-70), and PCC(NPA-60) were displayed in Figure 8a–f. Figure 8a shows that natural fiber is hydrophilic, and the time from the contact of water droplet with the fiber surface to complete penetration was 2.4 s. After alcohol-water-NaOH pretreatment, the proportion of the amorphous zone and the quantity of reactive –OH of the fiber increased, greatly enhancing its hydrophilicity. For PC(EtOH-70) and PC(NPA-60), as shown in Figure 8b,c, the water droplet penetration time was reduced to 0.4 s, which was one-sixth of the time required for the raw fiber. Furthermore, the introduction of hydrophilic quaternary ammonium groups onto the fiber by cationization can also improve the water absorption capacity of the fiber. The water droplet penetration time required for NCC (Figure 8d) was 0.8 s, which was much shorter than that for NC, but a little slower than that for PC(EtOH-70) and PC(NPA-60). As for PCC(EtOH-70) and PCC(NPA-60) shown in Figure 8e,f, the cationic samples of PC(EtOH-70) and PC(NPA-60), respectively, they possessed the best moisture absorption capacity, and water droplets were immediately absorbed as soon as they came into contact with the fiber surface.

### 3.6. Comparison of the Performance of Different Dyed Samples

Table 2 listed the comparison of exhaustion (*E*%), fixation (*F*%), K/S and color fastness of the dyed fibers obtained by various approaches. The table shows that the fixation and color depth of all salt-free dyed samples were higher than the results of conventional dyeing with salt. The traditional dyeing outcomes for natural fiber (NC) using 100 g/L of sodium sulfate and 20 g/L of sodium carbonate were as follows: *E*% = 79.8%, *F*% = 72.9%. For the salt-free dyeing of the raw-cationic fiber (NCC), when the dosage of GTA was 72.5 g/L, dye fixation of C.I. Reactive Black 5 was raised to 94.8%. For pretreated fibers, only 35 g/L or 30 g/L of GTA was needed for cationization to achieve high fixation salt-free dyeing with a dye fixation higher than 95.3%. Thus, pretreatment could reduce the quantity of cationic reagents required for fiber cationization by more than half.

Besides, Table 2 shows the color fastness of C.I. Reactive Black 5 on different cotton fibers. All rub and wash fastness reached 3–4 grade or higher, indicating good bonding properties between fibers and dyes. It is worth noting that wet rub fastness of the dye on PC(EtOH-70), PC(NPA-60), PC(IPA-60), and PC(TBA-60) all reached 4–5 grade, which is half grade higher than that on NCC, and one grade higher than that on unpretreated and uncationized NC. This may be due to the increased reactivity and permeability of the pretreated fabrics resulting in stronger bonding between dyes and fibers. The above results illustrated that the pretreatment had a positive effect on the cationization reaction and dyeing performance.

Figure 9a–d present the photographs of different dyed samples and the corresponding dyeing residues. Among them, Figure 9a is a dyed sample of NC, Figure 9b is a salt-free dyed sample of NCC, Figure 9c,d represent salt-free dyed samples of pretreated-cationic fiber: PCC(EtOH-70) and PCC(NPA-60), respectively. It can be seen that the color of the dyed samples turned to be darker from the left to the right, dyed PCC(NPA-60) had the darkest color, while the dyed NC showed the lightest one. For the pretreated fibers, the color was more blackish in hue, while the unpretreated fabrics had a blueish hue. The color of the dyeing wastewater shown in the inserted photographs just presented the opposite order. The wastewater of NC with conventional dyeing method is extremely dark due to low dye fixation, while that of the pretreated cotton is much clear owing to high dye utilization.

Furthermore, to verify the effectiveness of this approach, the cationic reagent consumptions were compared with those of some former references and listed in Appendix A [12,43,44,45,46,47]. In the present work, the quantity of GTA required to achieve salt-free dyeing at 6% (o.w.f) dye dosage was 30 g/L, i.e., 0.09 g of reagents exhaustion per gram of fiber, and the dye fixation was 96.0%. For achieving salt-free dyeing with high dye dosage, the cationic reagent usage in the previous references was more than five times the reagent dosage required in this paper. Hence, the method studied in this paper could achieve not only a lower consumption of GTA than the process reported previously but also a high dye fixation of salt-free dyeing.

Finally, to prove that the pretreatment solution was recyclable, the ethanol-water-NaOH system was selected for six pretreatment cycle trials, and the resulting fibers were sequentially modified (GTA dosage = 35 g/L) and salt-free dyed, the findings of fixation for salt-free dyeing were exhibited in Figure 10. It can be observed from the figure that the dye fixation of the six trials was between 95.2% and 97.1%, suggesting that the cycling did not affect much the dye fixation, which implied that the recycling of the pretreatment solution was feasible in this study.

## 4. Conclusions

In this work, a novel alcohol-water-NaOH pretreatment method was designed for increasing reactivity of cotton fibers. With this method, further cationization with GTA became much more efficient, and excellent salt-free dyeing performance of C.I. Reactive Black 5 was realized. Under the optimal conditions, the amount of GTA required for the cationization of the pretreated fibers could be reduced by half compared with that required for the untreated ones. XRD and FT-IR tests exhibited that the fibers pretreated with an ethanol-water-NaOH solution remained crystal type of cellulose I with a decrease in crystallinity, while when ethanol was replaced by *n*-propanol, isopropanol or *tert*-butanol, the pretreated fibers transformed to cellulose II with an increase in accessibility. In addition, the fibers swelled and turned to be more hydrophilic after pretreatment, which were conductive to cationization and dyeing. It is found the color of the dyed cotton pretreated with alcohol-water-NaOH was deep with high color saturation, and all color fastness was satisfactory, especially that the wet rub fastness could reach 4–5 grade. More importantly, the pretreatment solution could be effectively cycled and dye fixation maintained at high values of more than 95.2%, indicating the feasibility of the proposed scheme.

## Figures and Tables

**Figure 1 polymers-14-05546-f001:**
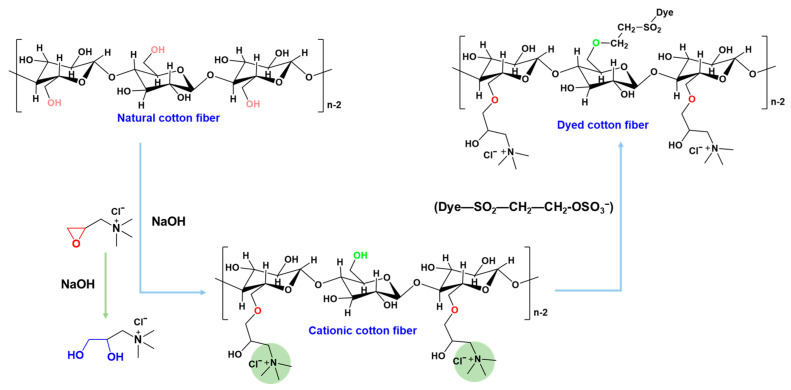
Schematic diagram of the reaction mechanism of GTA with natural cotton fiber and the reaction of reactive dye with the cationic cotton fiber.

**Figure 2 polymers-14-05546-f002:**
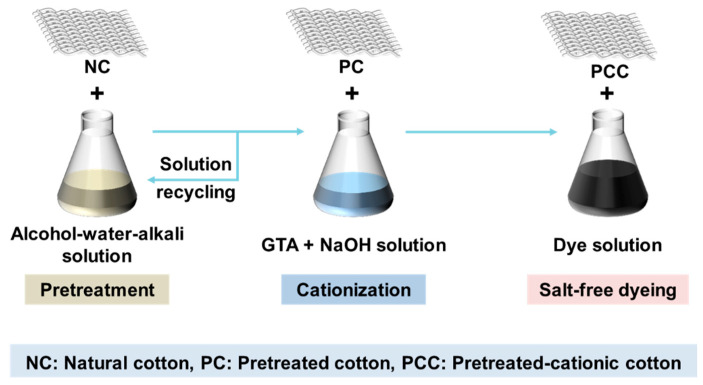
Schematic diagram of the pretreatment, cationization, and salt-free dyeing process of natural cotton fiber.

**Figure 3 polymers-14-05546-f003:**
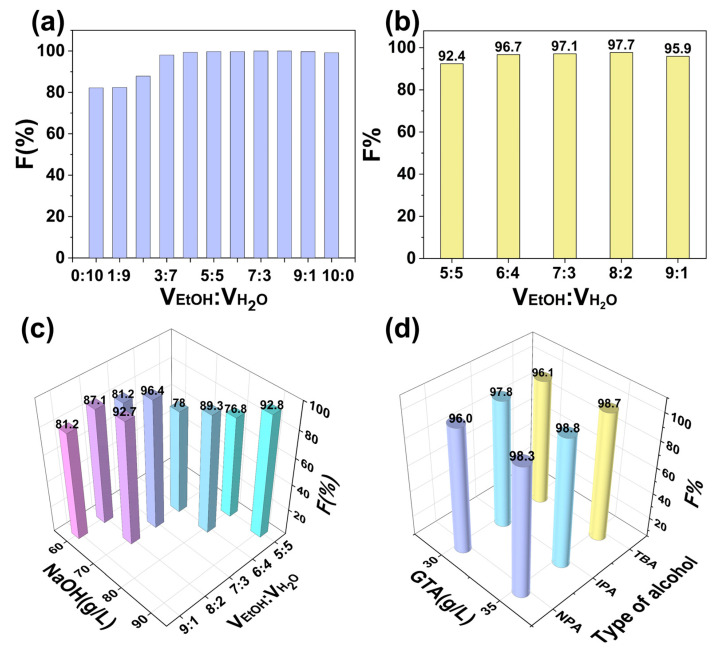
(**a**) Effect of V_EtOH_:$V_{H_{2}O}$ in ethanol-water-NaOH system on F% of C.I. Reactive Black 5 in salt-free dyeing, 55 g/L GTA was applied for the cationization of the pretreated fibers; (**b**) effect of V_EtOH_:$V_{H_{2}O}$ in ethanol-water-NaOH system on F% of salt-free dyeing, 35 g/L GTA was used during the cationization of pretreated fibers; (**c**) effect of NaOH quantity in pretreatment on F% of salt-free dyeing; (**d**) influence of the type of alcohol in pretreatment on F% of salt-free dyeing, where EtOH, NPA, IPA, and TBA represents ethanol, *n*-propanol, isopropanol, and *tert*-butyl alcohol, respectively, V_alcohol_:$V_{H_{2}O}$ was 8:2 and NaOH dosage was 70 g/L in ethanol-water-NaOH system or 60 g/L in the other three systems.

**Figure 4 polymers-14-05546-f004:**
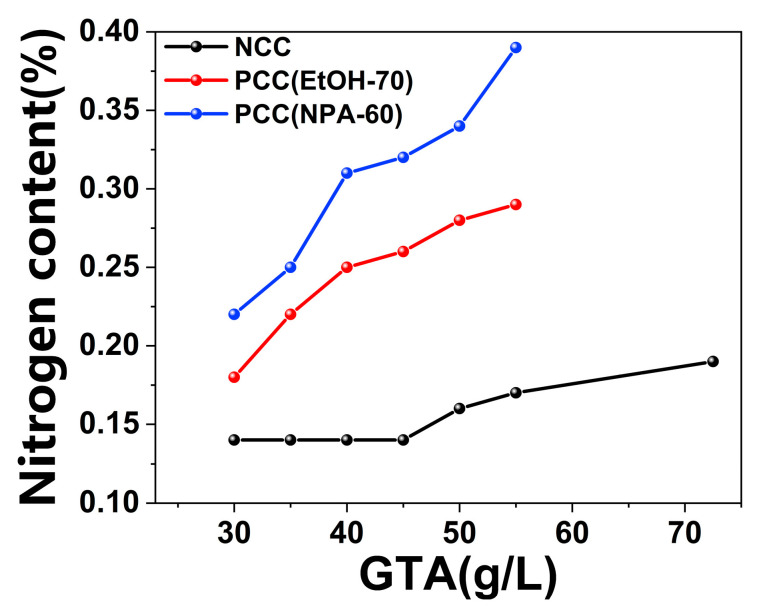
Nitrogen content of natural-cationic fiber (NCC), PCC(EtOH-70), and PCC(NPA-60) with different GTA dosage, the latter two were the cationic samples of PC(EtOH-70) and PC(NPA-60) which were pretreated with ethanol-water-NaOH (70 g/L) and *n*-propanol-water-NaOH (60 g/L).

**Figure 5 polymers-14-05546-f005:**
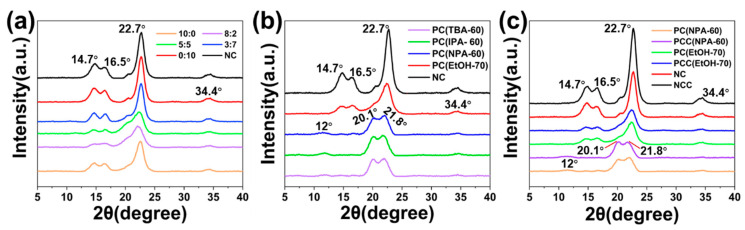
Comparison of XRD patterns of various cotton fibers: (**a**) natural cotton fiber (NC) and various PC(EtOH-100) which are pretreated using an ethanol-water-NaOH system with different V_EtOH_:$V_{H_{2}O}$, where NaOH usage was 100 g/L; (**b**) NC, PC(EtOH-70), PC(NPA-60), PC(IPA-60), and PC(TBA-60), the latter two pretreated fibers are obtained by pretreating with the alcohol-water-NaOH system of isopropanol, and *tert*-butyl alcohol, respectively, where V_alcohol_:$V_{H_{2}O}$ was 8:2 and NaOH dosage was 60 g/L; (**c**) NC, PC(EtOH-70), PC(NPA-60), NCC, PCC(EtOH-70), and PCC(NPA-60).

**Figure 6 polymers-14-05546-f006:**
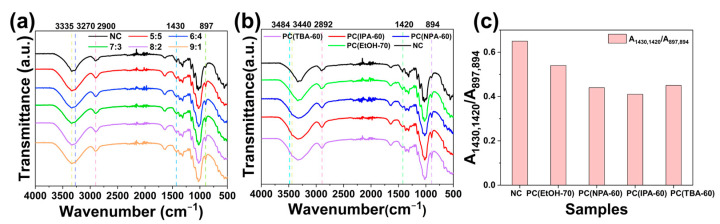
Comparison of FT-IR spectra of different cotton fibers: (**a**) NC and various PC(EtOH-100) with different V_EtOH_:$V_{H_{2}O}$; (**b**) NC, PC(EtOH-70), PC(NPA-60), PC(IPA-60), and PC(TBA-60); (**c**) FT-IR absorbance ratios (A_1430,1420_/A_897,894_) of NC and various pretreated fibers including PC(EtOH-70), PC(NPA-60), PC(IPA-60) and PC(TBA-60), respectively.

**Figure 7 polymers-14-05546-f007:**
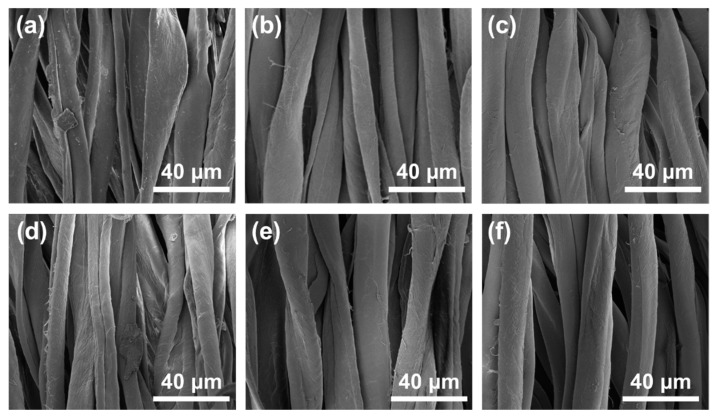
SEM images of different cotton fabrics with 2500× magnification: (**a**) NC, (**b**) PC(EtOH-70), (**c**) PC(NPA-60), (**d**) NCC, (**e**) PCC(EtOH-70), and (**f**) PCC(NPA-60).

**Figure 8 polymers-14-05546-f008:**
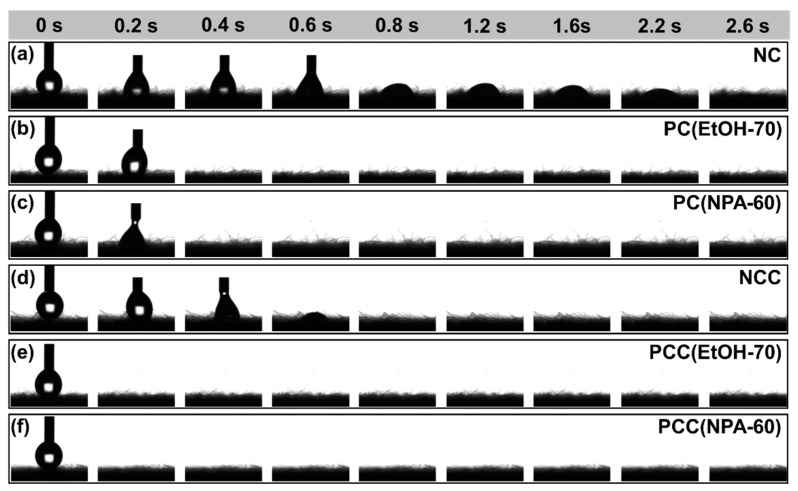
Comparison of hydrophilic properties of different fibers: (**a**) NC, (**b**) PC(EtOH-70), (**c**) PC(NPA-60), (**d**) NCC, (**e**) PCC(EtOH), (**f**) PCC(NPA-60).

**Figure 9 polymers-14-05546-f009:**
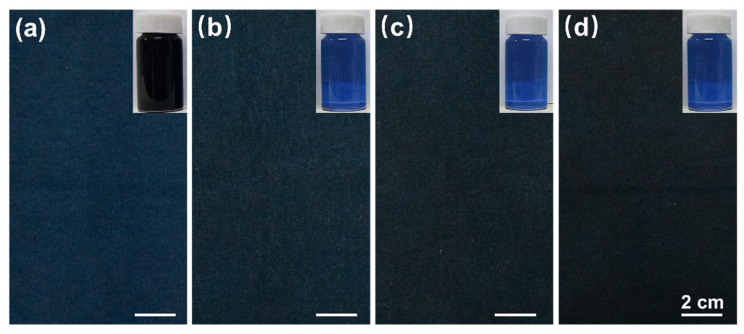
Comparison of the different dyed samples: (**a**) NC, (**b**) NCC, (**c**) PCC(EtOH-70), and (**d**) PCC(NPA-60), and the corresponding dyeing residues.

**Figure 10 polymers-14-05546-f010:**
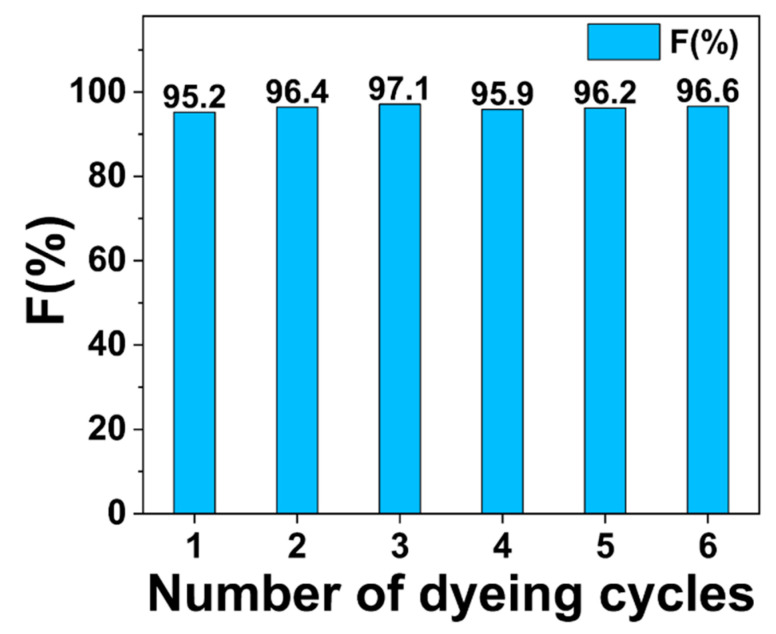
Recycling performance of ethanol-water-NaOH system, where V_EtOH_:$V_{H_{2}O}$ = 8:2, NaOH dosage = 70 g/L.

**Table 1 polymers-14-05546-t001:** Crystallinity of natural fiber and different pretreated fibers.

Samples	Crystallinity
NC	72.3%
PC(EtOH-70)	61.2%
PC(NPA-60)	54.6%
PC(IPA-60)	54.2%
PC(TBA-60)	54.9%

**Table 2 polymers-14-05546-t002:** Dye exhaustion, dye fixation, K/S, and color fastness of cotton fibers produced by different dyeing methods.

Samples	GTA Dosage (g/L)	*E*%	*F*%	K/S	Rub Fastness		Wash Fastness		
					Dry	Wet	Color Change	Staining on Cotton	Staining on Wool
NC	×	79.8%	72.9%	35.5	4–5	3–4	4–5	4–5	4–5
NCC	72.5	95.1%	94.8%	38.4	5	4	4–5	4–5	4–5
PCC(EtOH)	35	95.9%	95.3%	38.7	4–5	4–5	4–5	4–5	4–5
PCC(NPA)	30	96.5%	96.0%	39.3	4–5	4–5	4–5	4–5	4–5
PCC(IPA)	30	98.0%	97.8%	39.3	4–5	4–5	4–5	4–5	4–5
PCC(TBA)	30	96.6%	96.1%	39.3	4–5	4–5	4	4–5	4–5

## Data Availability

All the date is available within the manuscript.

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
