# Peer review of "Efficient Cationization of Cotton for Salt-Free Dyeing by Adjusting Fiber Crystallinity through Alcohol-Water-NaOH Pretreatment"

_polymers, 2022, doi:10.3390/polym14245546_

Round 1

Reviewer 1 Report

Aini Wu et al. in their manuscript they consider an interesting and topical topic of coloring cellulose fibers. The authors propose to use the alcohol-water-NaOH system for the pre-treatment of cotton fibers (Bleached, desized, and mercerized cotton fabrics). The question immediately arises, what is the chemical composition of cotton? How much fat does cotton contain? Why was cotton chosen over viscose fibers? Figure 3a. Why is there a decrease in F? What is the difference between (a) and (b)? 3.2. XRD analysis of Cotton Fibers. It is necessary to provide data on the crystallinity of cellulose! Lines 304-312. As I understand it, in this case, cellulose is dissolving, hence changes in the structure (on diffraction patterns) should appear?! Line 325. What is the concentration of NaOH in the system? 3.3. FT-IR Analysis of Cotton Fibers. It is necessary to match the data for crystallinity with the XRD results. Line 422. "Water contact angle is an indication of the water absorption capacity." - you need to add a link. Figure 8. Different fibrillar surface observed for the samples?!

The presented method requires various reagents, as it is supposed to regenerate these complex systems?

Reviewer 2 Report

This is an interesting study in that it developed a process for dyeing cotton into dark colors without using salt, which is used in the cotton dyeing process (and has a high environmental impact). However, the interpretation of the results of the analysis conducted to elucidate the mechanism requires reexamination. I list my comments below. If they are appropriately answered, I will judge that this paper is worthy of publication.

1.

In the introduction, the authors state that conventional mercerization is difficult to make a practical application because of alkali consumption facing the significant ecological stress. Nevertheless, is it not self-contradictory to use NaOH in this study? Also, mercerized cotton fabrics are used as raw materials, but is it essential that the cotton fabrics be mercerized?

2.

In this treatment, a pretreatment is performed to reduce the crystallinity of the cellulose, which facilitates the penetration of the dye or cationizing agent. As a result, it is possible to dye cotton fabrics in dark colors. On the other hand, there is concern about whether the strength of the cotton cloth after dyeing is maintained at a level that is acceptable for practical use. Have the authors verified this point?

3.

I think the environmental impact of EtOH and other alcohols as well as salt could be a problem. It would be good to have a statement regarding them.

4.

In the description of Figure 3(a) and (b) on page 5, it is stated that F% gradually increases with increasing VEtOH:VH2O. However, the F% vs. VEtOH:VH2O plot appears to show a change only at certain volume ratios. It is necessary to accurately represent the changes seen in the figure.

Also, Fig. 3(b) is plotted as a bar graph while Fig. 3(a) is plotted as a line graph. Although the two graphs differ in the amount of GTA added, the same physical properties are shown on both vertical and horizontal axes, so it would be easier to compare the two graphs if they were drawn on the same type of graph. It is better to show F% from 0% as in Fig. 3(c) and (d) to understand the situation more accurately.

5.

If NaOH is dissolved in a mixture of alcohol and water, will a basic metal alkoxide (RO-Na+) be formed? The higher the number of carbons in the alkyl chain, the more basic the metal alkoxide will be, so this point should be considered in addition to solubility.

6.

p.8

The authors describe it as follows:" This was because n-propanol, isopropanol, and tert-butyl alcohol had a weak capacity to dissolve NaOH, almost all NaOH was dissolved in water and a high concentration alkali solution was formed."

Based on the solubility of each alcohol in water, it appears that there is no phase separation in this system. If it is one phase, this part of the description is not reasonable.

7.

In the FTIR spectrum shown in Figure 6, are the hydrogen bonded hydroxyl group peaks (3335, 3270, 3484 and 3440 cm-1) identified by peak separation? The very broad and strong OH stretching absorption makes it difficult to recognize them in the data shown.

8.

The first paragraph of 3.6 describes the entire contents of Table 1. Since these contents are obvious from the Table, it is better to summarized the main points.

Round 2

Reviewer 1 Report

The authors answered the questions posed, made appropriate amendments to the manuscript. Work can be considered for publication.

Reviewer 2 Report

The authors responded courteously to my comments, and they have made appropriate additions and corrections to the manuscript and supplementary information, which I judge to be ACCEPTABLE.